# Modified Leach Residues from Processing Deep-Sea Nodules as Effective Heavy Metals Adsorbents

**Nguyen Hong Vu [1],\*, Eva Kristianová [1], Petr Dvořák [1], Tomasz Abramowski [2], Ivo Dreiseitl [2] and Aigerim Adrysheva [1]**

[1] Department of Metals and Corrosion Engineering, University of Chemistry and Technology Prague, Technická 5, 16628 Prague, Czech Republic; kristiae@vscht.cz (E.K.), dvorakp@vscht.cz (P.D.), adryshea@vscht.cz (A.A.)

[2] Interoceanmetal Joint Organization IOM, Cyryla i Metodego 9, 71-541 Szczecin, Poland; t.abramowski@iom.gov.pl (T.A), i.dreiseitl@iom.gov.pl (I.D.)

\* Correspondence: vun@vscht.cz; Tel.: +420-220-445-025

**Abstract:** The possible use of leaching residue from leaching deep-sea nodules in $SO_2/H_2SO_4/H_2O$ medium as a low-cost adsorbent of heavy metals (Pb(II), Cd(II), Cu(II), Ni(II), Co(II), As(V)) was studied. The leaching residue was found to be an effective adsorbent for all of the tested elements; however, it was inactive in the solution containing As(V). The chemical activation of adsorbent in 10 vol. % HCl resulted in the greatest improvement of adsorption properties, while the activation in 10 vol. % $HNO_3$ and heat treatment at 250 °C did not significantly affect the sorption characteristics of treated adsorbents compared with the original leaching residue. After HCl activation, the maximal adsorption capacities for lead (12.0 mg/g at pH 5.0 after 1 h), nickel (3.1 mg/g at pH 5.5 after 4 h) and cobalt (2.0 mg/g at pH 5.0 after 2 h) were achieved. Additional mechanical treatment connected with HCl activation provided the highest adsorption capacities for cadmium (11.5 mg/g at pH 4.0) and copper (5.7 mg/g at pH 4.5). Coprecipitation of Fe/Al-based particles on the surface of the leaching residue increased As(V) removal of the adsorbent. Surface coating based on $Al^{III}$ was extremely effective, causing the increase of the adsorption capacity from 0 with the original leaching residue, to 28.1 mg/g (pH 7.0, 24 min). Kinetics studies showed the rapid progress of adsorption for Pb(II), cadmium(II), and As(V) in tens of minutes, while the adsorption of Cu(II), nickel(II) and Co(II) approached a steady state after 2 h.

**Keywords:** deep-sea nodules; leaching residue; adsorption; heavy metals

## 1. Introduction

Deep-sea manganese polymetallic nodules are sedimentary formations covering the bottom of many global seas and oceans. They represent a potentially important resource of strategic metals such as Ni, Cu, Co, Mn and Fe. However, the large variety of other present metals (Mo, Zn, Zr, Li, Ti, Ge, platinum group metals and rare earth elements) increases the attractiveness of nodules for industrial processing. The nodules consist mainly of amorphous oxides and oxyhydroxides of Mn, Fe, Si and Al where Fe and Mn are present in variable oxidation states [1,2].

A number of technologies have been developed for nodules treatment [3]. Hydrometallurgical processes mostly use acidic or ammoniacal reduction leaching when Cu, Ni, Co and partly Mn and Fe transfer into a solution, leaving behind leaching residue that, after additional extraction of Mn and Fe, becomes waste. Proper utilization of solid leaching residues is therefore needed to enhance the proposed technologies from the economical and environmental point of view. Nodules are characterized by high porosity and surface area, the properties that were found to be preserved or

enhanced even after leaching, and therefore leaching residue is suitable as an adsorbent [4–6]. Its use in the treatment of industrial wastewaters containing various heavy metals might cope with conventional methods for water purification such as chemical reduction, ion exchange, reverse osmosis, coagulation, complexation, electrodeposition, solvent extraction, electrochemical treatment and adsorption on activated charcoal [7–15].

Studies on adsorption of heavy metals on deep-sea nodule leach residues have been carried out mainly in India. The residues generated by $NH_3$-$SO_2$ leaching of manganese nodules were successfully used as an adsorbent of $Ni^{2+}$, $Cr^{6+}$, $Cu^{2+}$, $Cd^{2+}$ and $Pb^{2+}$ [4,5,16–18], with removal efficiency strongly dependent on pH, temperature and usually initial metal ion concentration in the solution. Randhawa et al. found that manganese nodule residue obtained from reduction roasting followed by ammoniacal leaching is an acceptable adsorbent of $Ni^{2+}$ with good regenerability [19]. The leach residues were also used as an effective selenite adsorbent with the best adsorption capacity at pH 5 [20,21]. Besides metals, the adsorption of phenolic compounds [22], phosphate [23,24] and dye remains [25] onto leaching residue was reported as a possible alternative of their removal from industrial effluents. In previous studies, the leached residues mainly contained Mn, Fe, Si and Al as a result of selective extraction of Ni, Co and Cu from nodules by leaching.

Researches show that effective composite adsorbents can be synthesized by forming an oxide surface coating on another supporting material, and that these composite adsorbents can be easily separated from the aqueous solution after an adsorption process. Oxides were found to have the ability to remove metals to trace concentrations, and the adsorbed metals can be recovered and reused [5]. For that purpose, Fe and Mn oxides are especially used [26,27]. Nowadays, there is an increasing tendency to employ nanostructured materials in many fields, including the removal of heavy metals from wastewater effluents, due to their unique physical, chemical and mechanical properties [28]. In comparison to microstructured adsorbents, the nanostructured adsorbents have several advantages, such as higher surface to volume ratio, or smaller quantities of produced secondary pollutants [29]. Singh et al. found that $\alpha$-$MnO_2$ nanorods and $\delta$-$MnO_2$ nanofibers have a high adsorption capacity for arsenite [29]. Pectin-iron oxide magnetic nanocomposite was used as an adsorbent for the removal of Cu (II) from aqueous solution [30]. Zinc (II) oxide nanorods loaded on activated carbon showed very good results in the adsorption of heavy metals ($Cd^{2+}$, $Co^{2+}$) and dye residues (methylene blue, crystal violet) [31].

The objective of this research was to prepare, characterize and describe the adsorption potential of materials based on leaching residues obtained from deep-sea nodule leaching in $SO_2$-$H_2SO_4$-$H_2O$ medium. Leaching residue modification by chemical, mechanochemical and heat treatment was carried out in order to create effective and low-cost adsorbents of heavy metals. Moreover, the preparation of new nanocoated adsorbents based on the leaching residues, and the efficiency of their subsequent use for As(V) removal, was studied.

## 2. Materials and Methods

### 2.1. Preparation of Adsorbents

The deep-sea nodules were obtained from Interoceanmetal Joint Organization, Szczecin, Poland. The content of their main constituents is shown in Table 1.

**Table 1.** Main constituents of used nodules.

| Element | Mn | Fe | Si | Al | Mg | Ca | Na | Cu | Ni | Ti | Zn | Co |
|---------|-----|------|------|------|------|------|------|------|------|------|------|------|
| wt. % | 30.57 | 4.41 | 3.53 | 2.16 | 1.87 | 1.84 | 1.64 | 1.18 | 1.14 | 0.35 | 0.14 | 0.13 |

Nodules were repeatedly washed with portable and distilled water to get rid of soluble ions and to displace seawater from nodules pores. The washed nodules were dried at 70 °C for 3 days and ground in a vibration laboratory mill for 10 min. Based on our previous work [2], the leaching of nodules was carried out in a solution containing 20 g/L $SO_2$ and 100 g/L $H_2SO_4$ at 90 °C for 60 min with a liquid to solid ratio = 10:1. After filtration, leach residue was washed several times with

distilled water to remove all the sulfate, then dried in an oven at 100 °C for 24 h. The following adsorbents were prepared by the leaching-residue treatments described below.

Adsorbent A1: Leaching residue was used as prepared without additional processing.

Adsorbents A2/Cl and A2/N: Leaching residue was added to a solution of 10 vol. % HCl and $HNO_3$, respectively (liquid to solid ratio of 20:1). The suspension was agitated for 24 h at laboratory temperature. The adsorbents were repeatedly washed with distilled water and dried at 100 °C for 12 h.

Adsorbent A3/1 and A3/Cl: Adsorbent A1 and A2/Cl were heated in an oven at 250 °C for 8 h.

Adsorbent A4/Cl: Adsorbent A2/Cl was milled in Planetary Micro Mill PULVERISETTE 7 (Fritsch Ltd., Idar-Oberstein, Germany) using water medium and operating at 600 rpm for 30 min. The product was washed with ethanol and dried in the air.

Adsorbent A5/Fe$^{II}$: 1 g of leaching residue and 50 mL of 0.6 mol/L $FeSO_4 \cdot 7H_2O$ prepared under $N_2$ atmosphere was agitated at 250 rpm in a sealed polyethylene bottle at laboratory temperature for 24 h. The solid phase was filtered off, washed with distilled water, dried at 50 °C for 12 h and homogenized.

Adsorbent A5/Fe$^{III}$: 1 g of leaching residue with 50 mL of 0.025 mol/L $Fe(NO_3)_3 \cdot 9H_2O$ and 0.05 mol/L NaOH were agitated at 250 rpm in a sealed polyethylene bottle at laboratory temperature for 24 h. Then the solid phase was filtered off, washed with distilled water, dried at 50 °C for 12 h and homogenized.

Adsorbent A5/Al$^{III}$: NaOH solution (4 mol/L) was gradually added into 250 mL of 1 mol/L $AlCl_3 \cdot 6H_2O$ up to pH about 6.0, until the formation of AlOOH precipitate, then 20 g of leaching residue was mixed with the suspension of AlOOH and agitated in a sealed polyethylene bottle at laboratory temperature for 24 h. The solid phase was filtered off, washed with distilled water, dried at 50 °C for 12 h and homogenized.

## 2.2. Sorbents Characterization

Chemical and phase composition analysis of adsorbents was performed using an X-ray fluorescence spectrometer (XRF) ARL 9400 XP (Thermo Fisher, Ecublens, Switzerland) and an X-ray diffraction spectrometer (XRD) PANalytical X'pertPRO (PANalytical, Almeo, Netherlands). Structural characterization was carried out using a scanning electron microscope (SEM) TESCAN Vega 3 LMU (Tescan, Brno, Czech Republic). Size and size distribution of adsorbent particles were determined by Partica LA 950/V2 (HORIBA Scientific, Kyoto, Japan), and laser-particle sizers Analysette 22 MicroTec plus (Fritsch Ltd., Idar-Oberstein, Germany). The average specific surface area was measured by the Brunauer–Elmet–Teller nitrogen adsorption method (BET–$N_2$ adsorption) along with the pore size and pore volume using an ASAP 2020 (Accelerated Surface Area and Porosimetry Analyzer; Micromeritics, Norcross, GA, USA). Thermogravimetry (TG) and differential thermal analysis (DTA) was performed by TG-DTA Setsys Evolution (Setaram, Caluire, France).

## 2.3. Determination of Zero Charge Point (pH$_{pzc}$)

For determination of zero charge point pH$_{pzc}$, an acid-base titration was carried out using 0.5 and 1.5 g of adsorbent suspended in 50 mL of 0.03 mol/L $KNO_3$ solutions and agitated for 24 h in an orbital shaker at 250 rpm until the pH was kept constant. Then, 0.1 mL of 0.1 mol/L KOH solution was added to the suspensions to alkalize surface sites, and suspensions were titrated by adding 0.05 mL of 0.1 mol/L $HNO_3$ solution under continuous agitation. After each addition of a defined amount of $HNO_3$, the corresponding pH value was recorded, and the plots of pH as a function of the added amount of acid were constructed for both samples. pH$_{pzc}$ was identified as the intersection point of the curves with the plot for a blank solution.

## 2.4. Adsorption Tests

Adsorption of Pb(II), Cd(II), Ni(II), Cu(II), Co(II) and As(V) were studied in model solutions prepared from $Pb(NO_3)_2$, $Cd(NO_3)_2 \cdot 4H_2O$, $CuSO_4 \cdot 5H_2O$, $NiSO_4 \cdot 7H_2O$, $CoSO_4 \cdot 7H_2O$ and

$Na_2HAsO_4 \cdot 7H_2O$ (all of analytical grade). Batch experiments were carried out by mixing 0.1 g of the adsorbent with 50 mL of a testing solution containing 100 mg/L of a tested metal in a 100 mL Erlenmeyer flask. The suspension was then agitated in an orbital shaker at 200 rpm for up to 7 h. The effects of pH and adsorption kinetics were evaluated for adsorbents A1, A2/Cl and A2/N in test solutions of Pb, Cd, Cu, Ni and Co. Based on the results obtained, the efficiency of adsorbents after thermal and mechanical treatment (A3/1, A3/Cl, A4/Cl) was then tested in same solutions at one chosen pH. Kinetics of As(V) removal from the solution were investigated for adsorbents with nanosized oxyhydroxide coating A5/Fe$^{II}$, A5/Fe$^{III}$ and A5/Al$^{III}$ at pH 7 and for comparison also for the adsorbent A1. Concentrations of heavy metals in solution were determined by atomic absorption spectrometer GBC 932 plus (GBC Scientific Equipment Ltd., Dandenong, Australia), and in the case of As(V) by flameless graphite furnace atomic absorption spectrometer AAS 880 (Varian, Palo Alto, USA). pH meter Orion 525 (Thermo Fisher Scientific, Grand Island, NY, USA) was used for pH measurements.

The amount of metal per gram of sorbent q (mg/g) was calculated using Equation 1:

$$q = (c_0 - c_t)V/m \tag{1}$$

where $c_0$ and $c_t$ are initial and final concentrations of the selected metal (mg/L), $V$ is the volume of the solution (L) and $m$ is the mass of an adsorbent (g).

## 3. Results and Discussion

### 3.1. Characteristics of Prepared Sorbents

According to XRF, all sorbents had a very similar elemental composition (Table 2), with the main constituents being Si, Al, Ba, Ca and K. The adsorbents A5/Fe$^{II}$, A5/Fe$^{III}$ and A5/Al$^{III}$ were enriched by Fe and Al, respectively, indicating the formation of new compounds of Fe and Al.

**Table 2.** The chemical composition of adsorbents (wt. %).

| Element | Adsorbent | | | | | | | | |
|---|---|---|---|---|---|---|---|---|---|
| | A1 | A2/N | A2/Cl | A3/1 | A3/Cl | A4/Cl | A5/Fe$^{II}$ | A5/Fe$^{III}$ | A5/Al$^{III}$ |
| Si | 38.71 | 41.87 | 40.24 | 40.30 | 40.18 | 41.04 | 39.16 | 39.73 | 27.78 |
| Al | 5.95 | 6.73 | 6.28 | 6.01 | 6.19 | 6.15 | 6.33 | 6.17 | 13.77 |
| Fe | 1.66 | 1.74 | 1.51 | 1.72 | 1.52 | 1.50 | 3.07 | 4.89 | 1.18 |
| K | 2.39 | 2.48 | 2.36 | 2.50 | 2.41 | 2.34 | 2.37 | 2.48 | 3.21 |
| Ca | 1.42 | 1.34 | 1.29 | 1.35 | 1.28 | 1.28 | 1.40 | 1.39 | 1.14 |
| Na | 0.89 | 1.06 | 0.87 | 0.91 | 0.88 | 0.88 | 1.00 | 1.01 | 0.67 |
| Ba | 3.60 | 3.62 | 4.12 | 4.01 | 4.14 | 3.50 | 3.95 | 3.23 | 2.06 |
| Mg | 0.41 | 0.42 | 0.53 | 0.42 | 0.53 | 0.50 | 0.45 | 0.45 | 0.23 |
| S | 1.10 | 0.92 | 0.62 | 1.14 | 0.62 | 0.54 | 1.43 | 0.89 | 0.68 |
| Sr | 0.29 | 0.15 | 0.14 | 0.30 | 0.14 | 0.15 | 0.27 | 0.18 | 0.09 |
| Ti | 0.67 | 0.63 | 0.52 | 0.69 | 0.53 | 0.52 | 0.64 | 0.66 | 0.46 |
| Pb | 0.24 | 0.14 | 0.16 | 0.29 | 0.15 | 0.15 | 0.25 | 0.14 | 0.15 |
| Cl | 0.03 | 0.03 | 0.03 | 0.04 | 0.03 | 0.03 | 0.03 | 0.02 | 0.79 |

The XRD analysis shows that the original leaching residue consisted of six phases: albite ($NaAlSi_3O_8$), muscovite ($K(AlFe)_2AlSi_3O_{10}(OH)_2$), quartz ($SiO_2$), barite ($BaSO_4$), orthoclase ($K(AlSi_3O_8)$) and dickite ($Al_2Si_2O_5(OH)_4$) (Figure 1). This composition was preserved only after $HNO_3$ treatment. Other preparation methods lead to the elimination of orthoclase and dickite. Dickite was dissolved in HCl, probably due to an ion exchange reaction. Orthoclase reacts with HCl to form $HAlSi_3O_8$ and KCl. Otherwise, the phase composition of all adsorbents is similar, consisting of the four remaining phases and differing only in quantity of constituents (Table 3). No new crystalline phases containing Al or Fe were detected in the composition of the series A5 adsorbents, which can be explained by the formation of amorphous hydrous compounds such as hydroxides and/or oxyhydroxides.

The BET analysis confirms the assumption that all sorbents are very porous with a high specific surface over 200 m²/g (Table 4). The chemical treatment had almost no effect on the adsorbents' porosity. However, milling and heat treatment resulted in a slight reduction. With the deposition of Fe and Al onto the original leaching residue, porosity increased significantly. The increased specific surface of A5 adsorbents after Fe and especially Al deposition indicates the creation of new smaller particles.

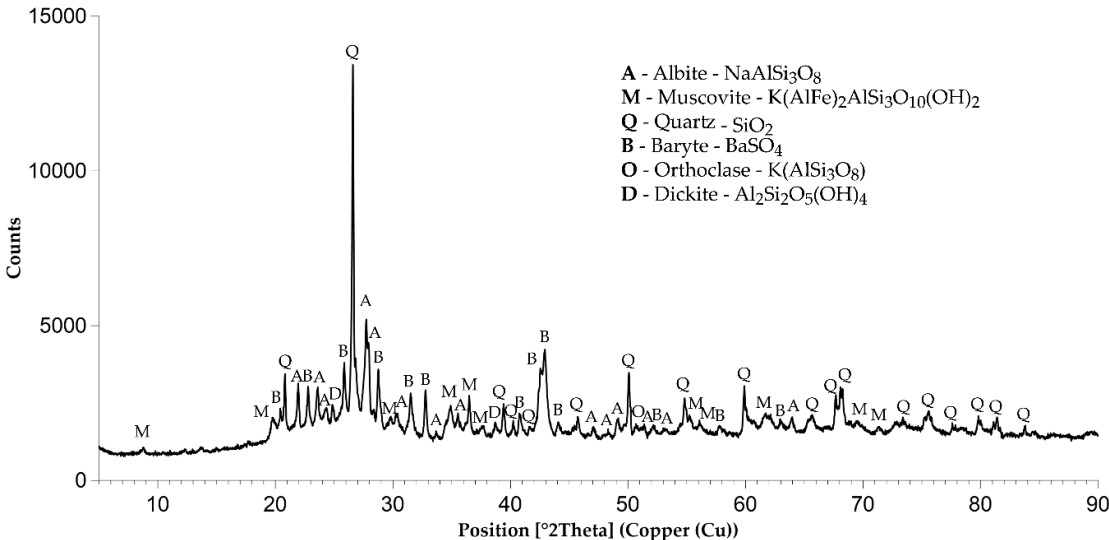

**A** - Albite - $NaAlSi_3O_8$
**M** - Muscovite - $K(AlFe)_2AlSi_3O_{10}(OH)_2$
**Q** - Quartz - $SiO_2$
**B** - Baryte - $BaSO_4$
**O** - Orthoclase - $K(AlSi_3O_8)$
**D** - Dickite - $Al_2Si_2O_5(OH)_4$

**Figure 1.** XRD pattern of adsorbent A1.

**Table 3.** Phase composition of the adsorbents.

| Phase | SemiQuant % | | | | | | | | |
|---|---|---|---|---|---|---|---|---|---|
| | A1 | A2/N | A2/Cl | A3/1 | A3/Cl | A4/Cl | A5/Fe^II | A5/Fe^III | A5/Al^III |
| Albite | 31 | 28 | 46 | 40 | 39 | 46 | 53 | 40 | 41 |
| Muscovite | 29 | 28 | 21 | 36 | 37 | 21 | 26 | 36 | 36 |
| Quartz | 19 | 19 | 27 | 19 | 18 | 27 | 16 | 20 | 19 |
| Barite | 5 | 6 | 6 | 5 | 6 | 6 | 5 | 4 | 4 |
| Orthoclase | 11 | 13 | - | - | - | - | - | - | - |
| Dickite | 5 | 6 | - | - | - | - | - | - | - |

**Table 4.** Specific surface of adsorbents (m²/g).

| Adsorbent | A1 | A2/N | A2/Cl | A3/1 | A3/Cl | A4/Cl | A5/Fe^II | A5/Fe^III | A5/Al^III |
|---|---|---|---|---|---|---|---|---|---|
| Specific surface | 234 | 227 | 229 | 220 | 211 | 216 | 274 | 253 | 298 |

The results of particle size distribution for the selected sorbents are presented in Figure 2. The heat treatment and chemical activation in HNO₃ of the original leaching residue (A3/1, A2/N) led to a negligible decrease in particle size, but slightly altered the size distribution. On the other hand, chemical activation in HCl had a positive impact on the reduction of bigger particles (A2/Cl, A3/Cl, A4/Cl). Subsequent heat treatment and milling of adsorbent A2/Cl reduced the amount of bigger particles even more, and significantly increased the size of smaller particle fractions. The particle size distribution of the series of A5 sorbents was similar to the distribution of the adsorbent A1.

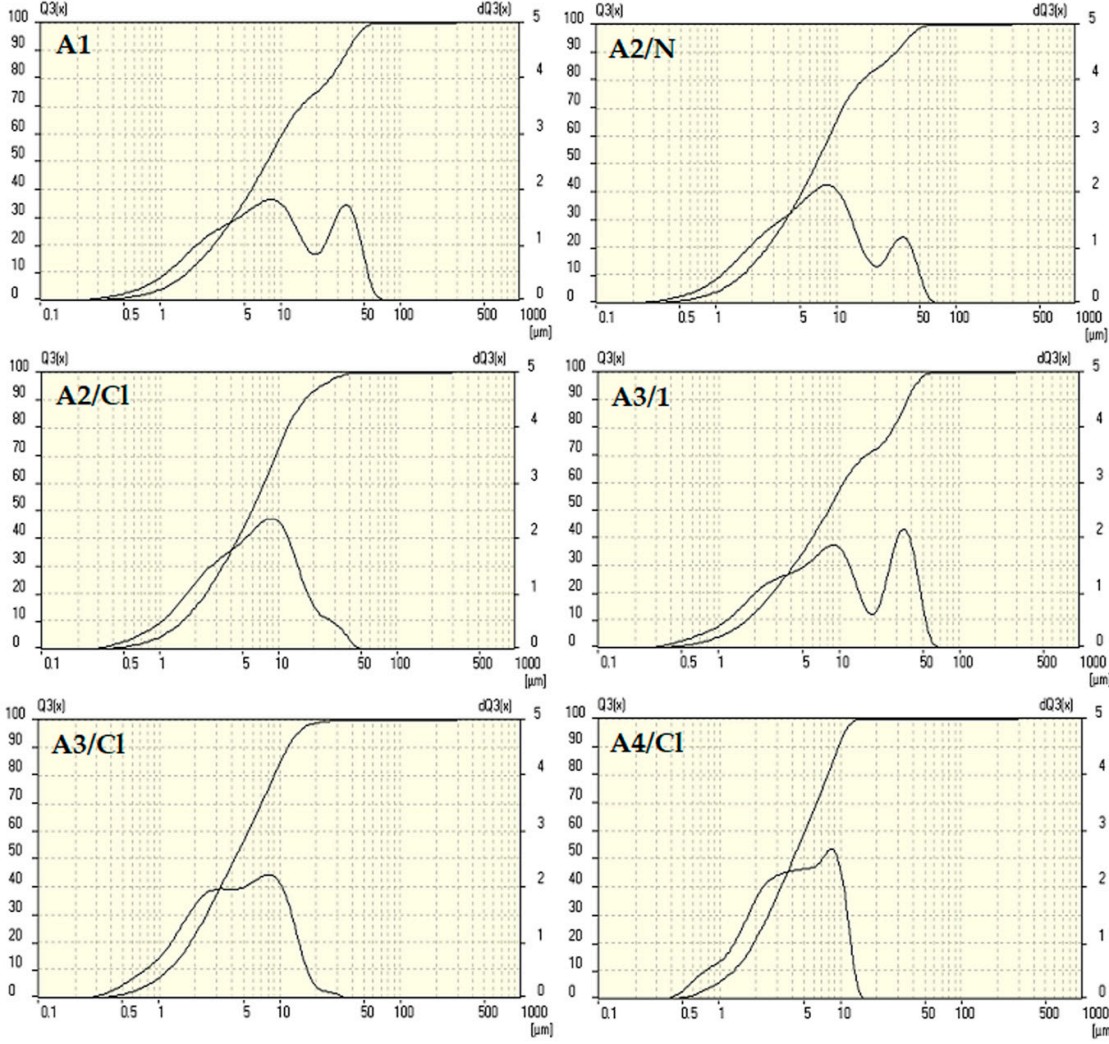

**Figure 2.** Particle size distribution of adsorbents.

SEM images of adsorbents show that adsorbents consisted of particles and aggregates of various particle sizes (Figures 3–4). The extensive aggregation of small particles resulted in high specific surfaces of the adsorbents (Table 5). The adsorbents A5/Fe$^{II}$, A5/Fe$^{III}$ and A5/A$^{III}$ (Figure 4) appeared to contain small particles of the particle size of approximately hundreds of nanometers deposited on much bigger particles of the original leaching residue. This deposition was the probable reason for the significant increase in specific adsorbent surfaces of the modified adsorbents.

Zero charge point (pH$_{pzc}$) of adsorbents A1, A2/N and A2/Cl was determined as 3.12, 3.05 and 2.96, respectively. At a pH higher than ~3, the surface of the adsorbent should gain a negative charge, which enables the effective binding of cations.

**Table 5.** Point of zero charge of selected adsorbents.

| Adsorbent | pHpzc |
|:---:|:---:|
| A1 | 3.12 |
| A2/N | 3.05 |
| A2/Cl | 2.96 |
| A3/1 | 3.10 |
| A3/Cl | 3.01 |
| A4/Cl | 2.97 |

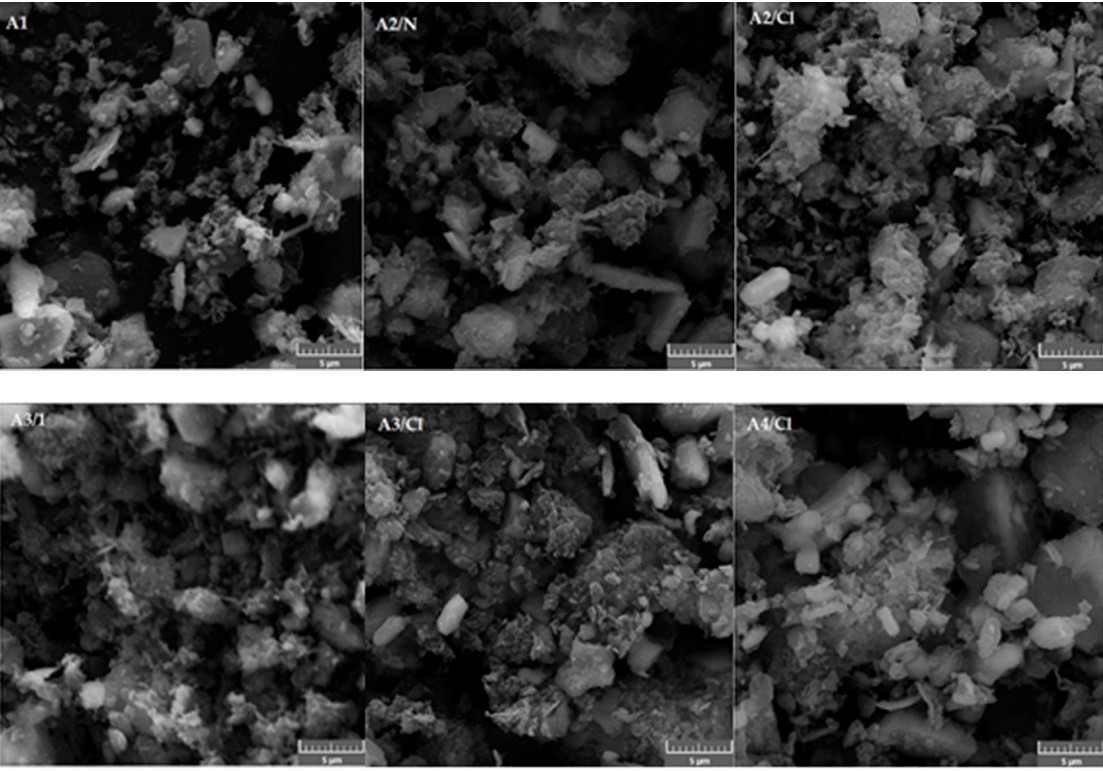

**Figure 3.** Electron images of adsorbents A1, A2/N, A2/Cl, A3/1, A3/Cl and A4/Cl.

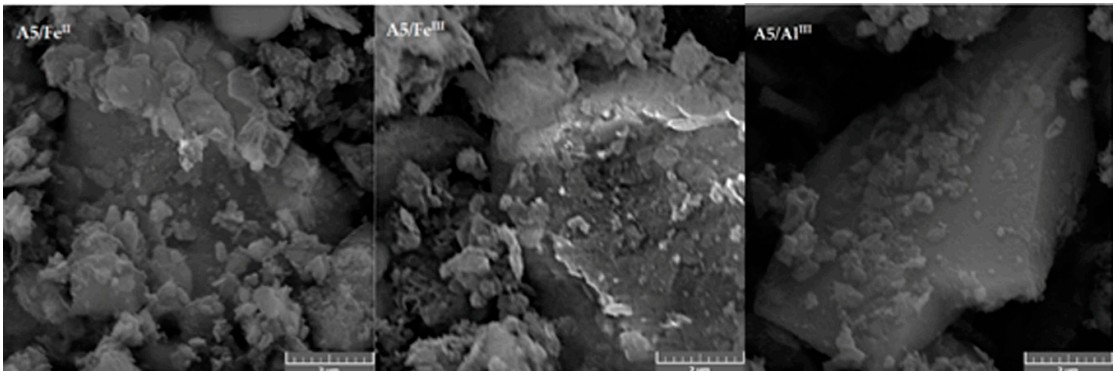

**Figure 4.** Electron images of adsorbents A5/Fe$^{II}$, A5/Fe$^{III}$ and A5/Al$^{III}$.

### 3.2. Adsorption on A1, A2/N and A2/Cl

The adsorption of Pb and Cd onto adsorbents A1, A2/Cl and A2/N proceeded very fast and completed in about 5 min (Figure 5a,b). At a pH higher than 2, the uptake of metals increased very fast and achieved the optimal value at pH 4 for Pb, and pH 5 for Cd. Higher pH led only to insignificant increases in adsorption capacities. Chemical activation of the adsorbent A1 in HCl did not affect sorption capacity for Cd, but led to increased adsorption capacity of Pb. After treatment in HNO$_3$, the adsorption capacities for both elements slightly decreased (Figure 6a,b).

The adsorption of Cu, Ni and Co was slow. Their sorption achieved a steady state after almost 4 h (Figure 5c–e). At a pH lower than 2.5, the adsorption did not proceed. Adsorption capacity grew rapidly from pH 2.5 to 3–4, then increased more slowly with increasing pH. Chemically activated adsorbents were more effective in the removal of Cu and Co, while the effectiveness for Ni removal was very similar for all three adsorbents (Figure 6c–e). Maximum adsorption capacities of A1, A2/Cl and A2/N for Cu, Ni and Co were lower than for Pb and Cd, but were still useful in practice.

The adsorbents A1, A2/Cl and A2/N were effective even at a pH lower than $pH_{pzc}$. A possible explanation is the combination of electrostatic adsorption mechanism and the ligand exchange mechanism, which might also contribute to the adsorption process.

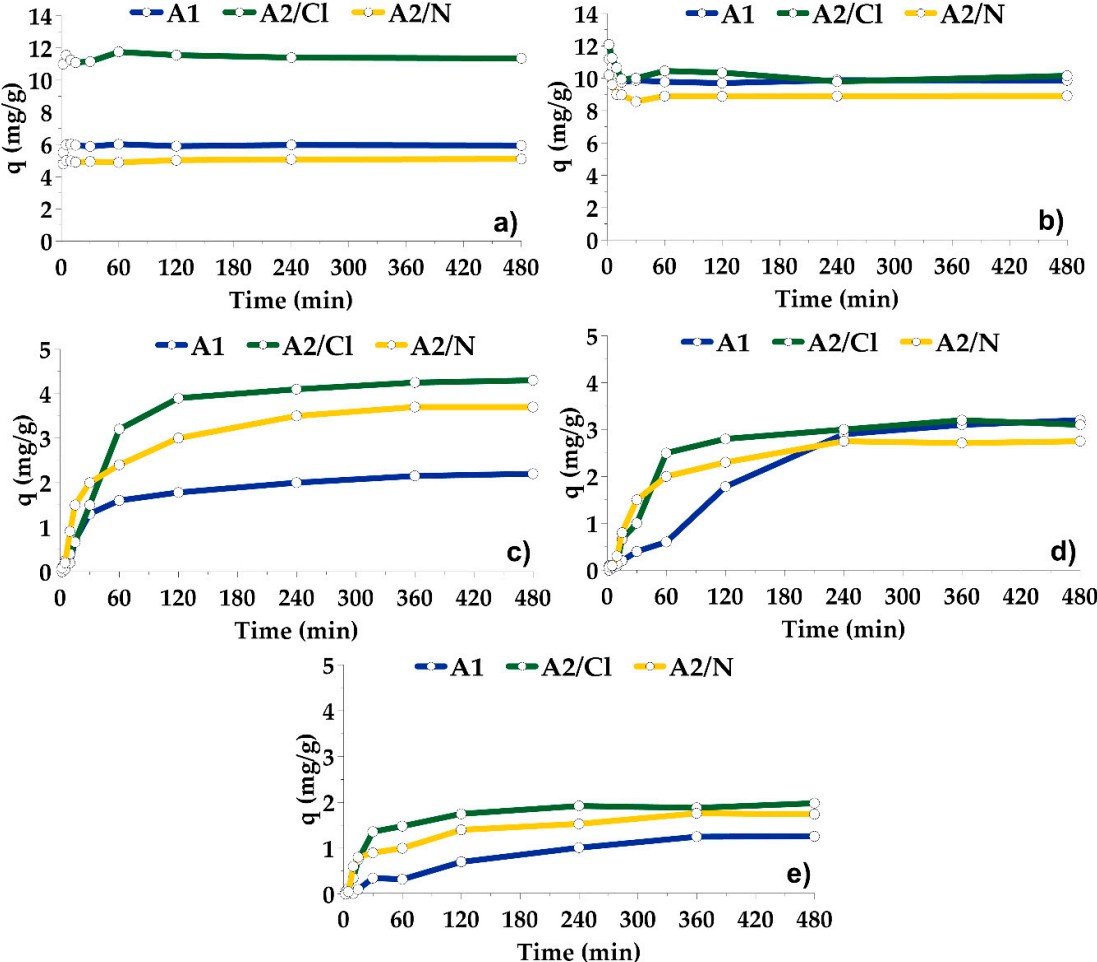

**Figure 5.** Adsorption kinetics of Pb (pH 5.0), Cd (pH 6.0), Cu (4.0), Ni (pH 5.5) and Co (pH 5.0) onto the adsorbents A1, A2/Cl and A2/N.

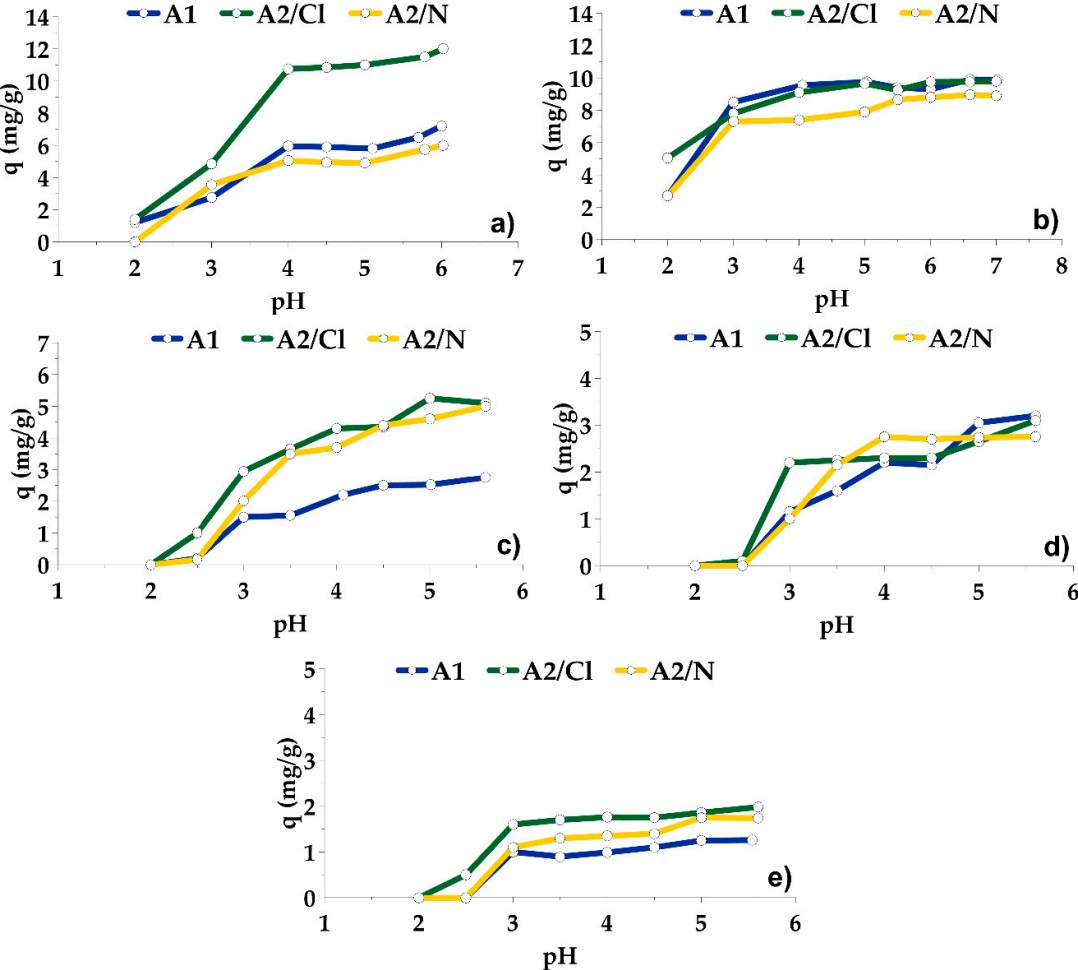

**Figure 6.** Effect of pH on the adsorption of (**a**) Pb, (**b**) Cd, (**c**) Cu, (**d**) Ni and (**e**) Co onto adsorbents after 8 h.

### 3.3. Adsorption on A3/1, A3/Cl and A4/Cl

Results of adsorption on thermally and mechanically treated adsorbents at selected pHs are presented in Table 6. The thermal treatment of adsorbent A1 was found to improve its adsorption capacity for all tested elements. Thermal treatment after the chemical activation in HCl, however, did not lead to any improvement in adsorption capacity. Mechanically treated adsorbent increased adsorption capacity only for Cd and Cu.

**Table 6.** The adsorption capacity of thermally and mechanically treated adsorbents after 8 h.

| Adsorbent | q (mg/g) | | | | |
|---|---|---|---|---|---|
| | **Pb** | **Cd** | **Cu** | **Ni** | **Co** |
| A1 | 5.0 | 9.8 | 2.5 | 2.2 | 1.1 |
| | pH 5.0 | pH 5.0 | pH 5.0 | pH 4.0 | pH 4.5 |
| A3/1 | 11.0 | 10.8 | 3.0 | 2.5 | 0.9 |
| | pH 5.0 | pH 5.0 | pH 5.0 | pH 4 | pH 4.5 |
| A3/Cl | 7.7 | 9.9 | 5.0 | 2.4 | 1.7 |
| | pH 5.0 | pH 5.0 | pH 5.0 | pH 4.0 | pH 4.5 |
| A4/Cl | 9.4 | 11.5 | 5.7 | 2.7 | 1.7 |
| | pH 4.5 | pH 4.0 | pH 4.5 | pH 4.0 | pH 4.5 |

Data from TG–DTA (Figure 7) show a big weight decease of about 8% around temperature 100 °C, indicating water release from the original leaching residue/adsorbent A1. After that, the weight loss increased gradually with increasing temperatures, indicating a probable release of chemically bound water at temperatures around 250 °C, and a gradual decomposition of adsorbent structure with increasing temperatures. This is presumably the reason why the thermal treatment was not effective in increasing the sorption capacity.

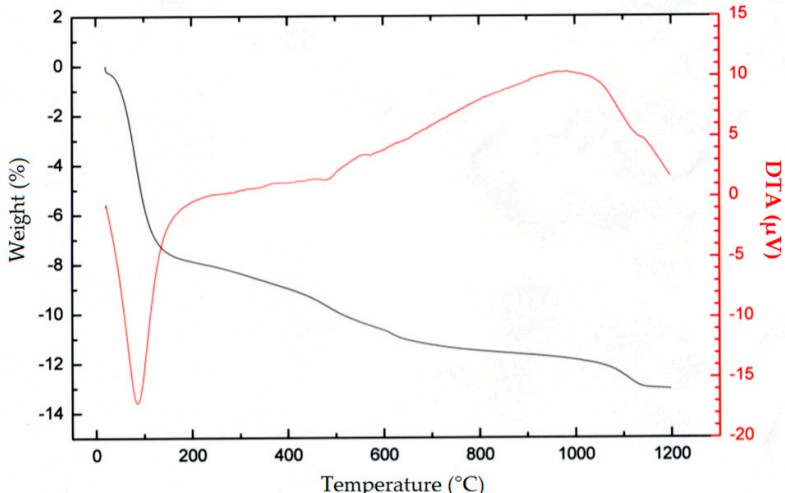

**Figure 7.** TG–DTA analysis of decomposition of the sorbent A1 in the air at heating rate 10 °C/min.

### 3.4. As(V) adsorption on A1, A5/Fe^{II}, A5/Fe^{III} and A5/Al^{III}

The adsorbent A5/Al^{III} was very effective in As(V) removal from the solution. It could adsorb a high amount of As(V) (28.1 mg/g) in a short time (~8 min) (Figure 8). Adsorbents with Fe coating had a significantly lower capacity (3.3 mg/g for A5/Fe^{II} and 1.5 mg/g for A5/Fe^{III}), but shared the equally quick adsorption kinetics. Adsorbent A1 was ineffective in As(V) removal from the solution, and therefore Fe/Al coating enables the utilization of leaching residue as adsorbent even for treatment of solutions containing this highly toxic metal.

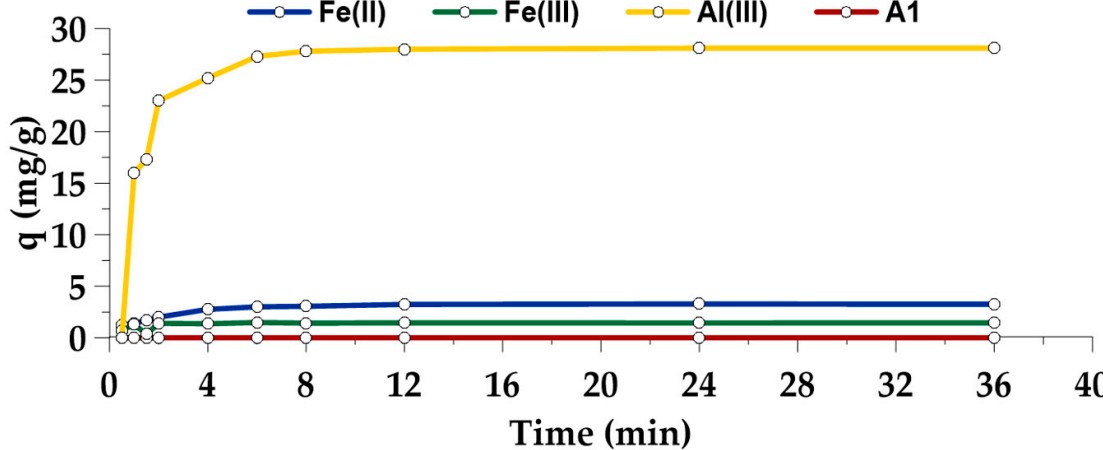

**Figure 8.** Adsorption kinetics of As(V) onto the adsorbents A5/Fe^{II}, A5/Fe^{III}, A5/Al^{III} and A1 at pH 7.0.

Compared to recently tested nanomaterials from various studies, the sorption abilities of newly prepared adsorbents with Fe coating were much lower; however, the adsorbent with Al^{III} coating appeared to have high potential to be one of the best materials from the group as a low-cost adsorbent of As(V). Adsorption characteristics of some studied sorbents are summarized in Table 7.

**Table 7.** Comparison of As(V) adsorption capacities of different adsorbents.

| Adsorbent | Initial pH | Adsorption capacity (mg/g) | Specific surface/particle size | Reference |
|---|---|---|---|---|
| Micro-/nanostructured Fe–Ni binary oxides | 7.0 | 90.1 | 245 $m^2/g$ | [32] |
| Iron oxide nanoparticle-coated single-wall carbon nanotubes | 4.0 | 49.7 | 2–3 nm | [33] |
| $Fe_3O_4$ nanoparticle-coated boron nitride nanotubes | 6.9 | 32.2 | 20–50 nm | [34] |
| $\alpha$-$MnO_2$ nanorods | 6.5 | 19.4 | 50–100 nm length, 5 nm wide | [29] |
| Ascorbic acid-coated $Fe_3O_4$ nanoparticles | 7.0 | 16.7 | 179 $m^2/g$ | [35] |
| Concrete/maghemite nanocomposites | 5.0 | 8.8 | nanopore 20–70 nm | [36] |
| $Al^{III}$-coated leaching residue | 7.0 | 28.1 | 298 $m^2/g$ | Present work |
| $Fe^{II}$-coated leaching residue | 7.0 | 3.3 | 274 $m^2/g$ | |
| $Fe^{III}$-coated leaching residue | 7.0 | 1.5 | 253 $m^2/g$ | |

## 5. Conclusions

The leaching residue from leaching deep-sea nodules in $SO_2/H_2SO_4/H_2O$ medium was utilized as an adsorbent for removal of heavy metal ions from aqueous solution. The series of adsorbents were prepared by chemical, mechanical and thermal activation, and coating with Fe and Al.

All adsorbents were found to have similar elemental and phase composition, with particle size ranging from a few hundred nm up to 100 µm, a high specific surface above 200 $m^2/g$ and a hygroscopic character.

The adsorption tests showed that untreated leaching residue and chemically activated leaching residues are very effective adsorbents of Pb(II) and Cd(II) in terms of high adsorption capacity and fast kinetics. Cu(II), Ni(II) and Co(II) can also be removed by those materials, but the adsorption capacity is lower, and the adsorption process is slower. The residue activated in HCl exhibited very good results for all tested elements, with adsorption capacities around 12 mg/g for Pb, 9.8 mg/g for Cd(II), 5.3 mg/g for Cu(II), 3.1 mg/g for Ni(II) and 2 mg/g for Co(II). The influence of heat and mechanical treatment on adsorption properties was mostly negligible. Mechanical activation increased the sorption capacity only in the case of sorption of Cd(II) and Cu(II).

The most-promising adsorbents were obtained by coprecipitation of particles based on $Fe^{II}$, $Fe^{III}$ and $Al^{III}$ onto the surface of leaching residue. These coated adsorbents were proven to be beneficial in the removal of As(V), especially the adsorbent based on $Al^{III}$ that exhibited significantly increased specific surface. While untreated leaching residue was not effective for As(V) removal, and Fe-coated adsorbents exhibited an adsorption capacity only around 3.3 mg/g ($Fe^{II}$) and 1.5 ($Fe^{III}$), the Al-coated adsorbent reached a final sorption capacity of approximately 28.1 mg/g in less than 10 min. Due to the easiness of the preparation procedure and high sorption capacity, the new Al-based adsorbent can be considered a very promising material for cheap As(V) removal, compared with the recent novel adsorbents, which are often prepared using complicated methods. Moreover, the utilization of waste leach residue as low-cost adsorbents for the effective removal of heavy metals will improve the economy and increase the effectiveness of technology for processing deep-sea nodules in a hydrometallurgical way, using $SO_2$ as the reductant in $H_2SO_4$ medium, as well as through methods for wastewater treatment involving adsorption.

**Author Contributions:** E.K. compiled the original draft and carried out experiments. H.N.V. worked as a scientific supervisor, designed the experiments, and evaluated results. P.D. helped with the measurements and constructed the apparatus. T.A. and I.D. secured the financial support and validated the results. A.A. revised the manuscript.

**Acknowledgments**: This research was financially supported by Interoceanmetal Joint Organization, Szczecin, Poland under international grant No. 106 19 0063.

**Conflicts of Interest:** The authors declare no conflict of interest.

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
