# Peer review of "Modified Leach Residues from Processing Deep-Sea Nodules as Effective Heavy Metals Adsorbents"

_metals, doi:10.3390/met9040472_

Reviewer 1 Report

Review of manuscript: metals-474394

This script describes the preparation of the heavy metal adsorbent from the leaching residue from processing deep-sea nodules, and discussion for the removal ability of the prepared adsorbents for heavy metal ions from aqueous solution. It is an interesting script, but it is not written properly. There are some points, which require revision and need to be clarified in the revised text. The points are described below.

1)      Abstract: l. 15, “for all the tested elements however, it is inactive” →“for all the tested elements, however, it is inactive”. l. 25, “in case of”→”for”.

2)       Abstract: l. 26, “reasonable level”. What is reasonable level? Water quality standard in Czech or Poland? You should write information for target level in detail in abstract and text.

3)      Introduction: l. 44-26, You should add the references for conventional methods.

4)      Materials and Methods: “FeSO4.7H2O”→“FeSO4‧7H2O”, ” Fe(NO3)3.9H2O”→” Fe(NO3)3‧9H2O”, ”AlCl3.6H2O”→”AlCl3‧6H2O”, “BET-N2 adsorption”→“BET-N2 adsorption”. Also, at first use in text, “XRD” and “XRF” are abbreviation, and you should use X-ray diffraction (XRD) and X-ray fluorescence (XRF).

5)      General: You use some terms for same materials, arsenic, As and arsenite. You should use As(V) because it is very important for As removal to use As(III) or As(V) (very different behavior).

6)      Results and Discussion: You should add the chemical formula for albite, muscovite, quartz, baraite, orthoclase and dickite. Also, you should indicate the figure of XRD.

7)      Results and Discussion: l. 173-179, “the original leaching residue lead to”→”the original leaching residue (A3/1, A2/N) lead to”, “the reduction of bigger particles.”→” the reduction of bigger particles (A2/Cl, A3/Cl, A4/Cl).”. You should add the figure for A5. Also, you should discuss what is dissolved using XRD and SEM data.

8)      Results and Discussion: l. 182-185, this figure indicate the reason why 250 oC heating is not effective. You should move this figure and paragraph into adsorption section.  

9)      Figure 3, 4: I can’t understand these figures and something you want to describe. You should revise the contents for these figures completely.

10)  Results and Discussion: l. 199-201, You should move this paragraph into adsorption section, and add the Table for pHpzc of all samples.  

11)  Figure 5-10: in order, “Figure 10”→”Figure 5”, and figures 5-9 is summarized into Fiuire 6 (a), (b), (c), (d) and (e). In figures, pH is initial pH or equilibrium pH? Also, pH of hydroxide formation for metal ions should be indicated because these heavy metals is easily precipitated as hydroxide by increasing pH.

12)  Graphs: You should draw the plots bigger.

13)  Table 4: You should add A1 data.

14)  Table 5: “α-MnO2”→ “α-MnO2”, “Fe3O4”→ “Fe3O4”

 I recommended publication of this paper, subject to the above points being satisfactorily addressed.

 Author Response

Dear Reviewer,

Thank you for your careful review, which has made the manuscript much better.

Pls. find attached the responses to the comments:

R1. text corrected.

R2. "reasonable level" changed to "approached to the steady state after 2 hours

R3. References added.

R4. Formulas and abbreviations were corrected

R5. As(V) is being used through out the manuscript

R6. Chemical formulas of phases added, X-Ray pattern of A1 added for illustration

R7. text corrected. Particle size of the series A5 was  explained in the manuscript. Dissolution of Dickite and Orthoclase was discussed (l. 163-168)

R8. Fig. was moved to Results and Discussion. 

R9. Figs 3,4 show that beside compact particles the adsorbents consist of a number of aggregates of nanoparticles. That it the main reason of high specific surface of the adsorbents. Fig 4 indicated deposition of nanoparticles of Al and Fe on the surface of the original adsorbent A1, which led to the increase of specific surface of the modified adsorbents of the series A5.

R10 We prefer to keep this text about pHpzc in Characteristics, where the main properties of adsorbents are described. 

R11. Figs. corrected as proposed. 

We are well aware of hydrolysis of the metals cations at high pH. On the other hand the solution is very diluted and no hydrolytic precipitation was observed during preparation of the solutions. It is probably that the precipitation takes much longer than sorption time in such diluted solutions.

R12. Graphs - corrected

R13. A1 data added

R14. Formulas corrected

Reviewer 2 Report

Line 13-What is the mineralogical composition of leaching residue?

Line 19- What is the concentration of HCl for the activation process?

Line 20- In which time is the maximal adsorption reached (1 day?)?

Line 21- In which time shall be this adsorption reached?

Line 23, In which form was Arsenic removed (Hydroxide)

Line 57.-by leaching (more details: pressure, temperature, s/L)

Line86- Which fraction is reached after grinding using a vibration laboratory mill for 10 min

Line 103-with 50 ml (not 50 m l)

Line 134- Did you all tests perform with synthetic solutions? Did you test your adsorbents in real waste metallurgical water?

Line 142- Nanosized oxyhydroxide coating (What is particle size, surface area,..)

Line 171, Table 3, Specific surface of adsorbents (m2/g). What is particle size of the adsorbent? Can you put the values of specific surface of other adsorbents from literature as a comparison

Line 176. Can you show the chemical reaction of the chemical activation with HCl

Line 181, As shown at  Figure 1. Particle size distribution of adsorbents, no nanoparticles below 100 nm, what is very important for the adsoption characteristics

Line 187.-The maximal weight loss amounts 14 %. Did you register some carbonate in this adsorbent structure

Line 261. Can you insert the information about surface area (particle size) in Table 5 for all absorbents? Which characteristics shall be compared in order to explain the favorable characteristics of your prepared adsorbent

Line 283. Due to the favorable characteristics,..Can you explain which characteristics are required for these favorable characteristics (chemical composition; morphology? Particle size? Core-shell structure? Porous structure?

 General comments?

Is core shell structure responsible for these favorable characteristics of the leaching residue

Can Modified leach residue from Processing Deep-Sea Nodules be used  as effective Rare earth elements adsorbents

What is the role of the Determination of zero charge point?

Can adsorbent be  used in scale up experiment of waste water treatment?

Can you prepare an adsorbent with controlled particle size and morphology?

Author Response

Dear Reviewer,

Thank you for your review and asked questions. Pls. find attached our responses to them.

R to L.13,19,20,21,23: The mineral composition of adsorbents were given in the manuscript (L. 163-184). Concentration of HCl, HNO3, adsoption times and sorption capacities were added to the abstract. The ionic form of arsenic was added as As (V) - arsenate.

R to L 57: the leaching conditions were given in the manuscript (L90-94)

R to 86: the particle size of the milled adsorbent A4/Cl and its morfology are shown in Fig. 2 and Fig. 3.

R to L 103: corrected

R to L134: all the tests were performed with model solutions. We are planning to test the real waste solutions in the future research.

R to L 142: the appearance of nanoparticles of the particle size of around hundreds nanometers can be observed in Fig. 4

R to L 171: Particle size distributions of adsorbents are illustrated in Fig. 2 and mentioned in the text (L192-193)

R to L 176" According to the obtained results and literature, only dickite and orthoclase dissolved during activation in HCl. Dickite can be dissolved in diluted HCl via ion exchange reaction (H.Ye. Matsuoka,Oleate flotation of dickite from quartz with diluted HCl preconditioning, IJMP, 40, 1-2, 99-109). Orthoclase can be dissolved in HCl to form HAlSi3O8 and KCl (H.D. Fogler and K. Lund, Acidization III - The kinetics of the dissolution of sodium and potassium feldspar in HF/Cl Acid mixtures, Chem. Eng. Sci, 30, 1975, 1325-1332)

R to L 181: SEM pictures of adsorbents (Figs 3,4) show aggregates of small particles of nano-scale size. It is probably the reason of high specific surface of the adsorbents. Even though the particle size distributions do not indicate the presence of nanoparticles in the adsorbents.

R to L 187: The leaching residues have high water content, which leads to high weight loss during sintering. Carbonate was not observed in the structure of the adsorbents.

R to L 261: Particle size and/or specific surface of listed nanosorbents were added to Tab 7 (previously Tab 5). The adsorbents prepared in the study exhibit high specific surface and amorphous nature od deposited oxyhydroxide of Al and Fe. These properties are the main reason for their effectiveness in As(V) removal.

R to L 283. Formulation was changed to "easiness of the preparation procedure and high sorption capacity". For removal of As(V) high specific surface and amorphous nature of aluminum oxyhydroxide are probably the main reasons for high As(V) removal.

Rs to general comments:

- We think high specific surface and chemical composition of the leaching nodules play the main role in their high sorption capacities.

-  We think it is possible since REEs have higher charge than divalent cations studied in our research.

- pHpzc indicates pH value above which the surface of a material changes from positive to negative. The negative surface attracts positive charges, leading to  sorption of cations.

- The method is easy to be applied and it can be scale-up.

- The leaching residues can not be changed but in theory a controlled deposition of nanoparticles on the leaching residues can be achieved by changing reaction conditions, eg. temperature, concentrations, pH, mixing rates.

Round  2

Reviewer 1 Report

There are a few corrections as follows.

Abstract, L.26: "in case" delete

General: You should use only As(V). arsenite, arsenic, As, etc. sholed be changed to As(V).

2.2 Sorbents characterization: you should add XRF and XRD, X-ray fluorescence sepectormeter (XRF), X-ray diffraction spectrometer (XRD). 

Author Response

Dear Reviewer,

We appreciate your  comments and suggestions. Pls. find our responses as followed:

R1, Abstract: corrected.

R2, Genera/As(V): corrected

R3, 2.2: corrected.

Reviewer 2 Report

Line 79 new nanocoated adsorbent (aimed particle size and purity? )

Line 286, 287-It is a probable reason that sorption capacity for  thermally treatment was not effective in increasing the sorption capacity. (Please to check this sentence/ Sorption capacity is two times mentioned)

Line 291- Adsorbents with (nano?)Fe coating

Line 295- containing this highly toxic metal such as ..(which metal? Pb?)

Line 300- Compared to recently tested nanomaterials (which type of materials?metals; oxides)

Line 329-a very promising material for a cheap arsenic, As(V) removal ("what is the Price of removal"; what is cheap: procedure or an adsorbent?

Line 332- low-cost adsorbents for effective removal of heavy metals (from waste water? in which reactors?) 

line 333-processing of deep-sea nodules by hydrometallurgical way (under high pressure in an autoclave? or under atmospheric pressure?

line 333-and the methods for waste water treatment. (which methods: precipitation, electrocoagulation)

General comments and questions:

1.  What is the final purity of the "Fe-based adsorbents!

2. I did not see any reactor for the testing of nanoadsorbent. CAn you give more Information (in your conclusion) about possible application of nanoadsorbent in a reactor?

3. Modified Leach Residues from Processing Deep-Sea Nodules are very Effective Heavy Metals Adsorbents. Can we use the prepared nanoadsorbent for the Sorption of heavy rare earth elements?

4. Can you calculate the kinetics of the Sorption process?

5. What is the selectivity of the sorption process? Can we control the Sorption selectivity via changes of the pH-values?

6. What is the influence of the temperature on the Sorption Efficiency?

Author Response

Dear Reviewer,

Here are our responses to your comments and/or questions:

R1/L79:  the aim of the study was to examine if the deposition of Fe nanocoating on the leaching residue was possible. This is a trial and error approach. We could not specify the size of coating we aimed to get  when we did not know whether it is even possible at first. It is equally true regarding the purity of the coating.

R2/L286-287(actual L248-249): the sentence rewritten for better understanding. 

R3/L291 (actual L285):  changed to Fe-coated adsorbents.

R4/L 295 (actual L256): It is clearly pointed to As in the context of the paragraph.

R5/L300 (actual L 288): It is the summary of results achieved, comparing to the adsorbents listed in Tab. 7. It is not necessary to relist them again since conclusions must be short and concise. 

R6/L329: It is cheap because of utilization of waste (leaching residues), easy to use (simple preparation) and effectiveness (high capacity). All of these arguments are clearly substantiated from the results and discussions. 

R7/L332: Yes, from waste water. We are not sure what reactors the Reviewer wanted to know in Conclusions. All procedures are mentioned in Experimentals.

R8/L333:In Introduction and Method the hydrometallurgical way for processing deep -sea nodules was described. For better understanding some info about the process was added.

R9/L L333: adsorption added.

Responses for General comments and questions:

R1. XRF confirmed the increase of Fe content in the Fe coated adsorbents". Regarding the coating, it is logical that it is of only Fe.

R2. The reactors eg. 100 mL Erlenmeyer flask was added to the Experimetal.

R3. We do not  think it is possible to use the modified adsorbents for REEs removal since REEs precipitate at pH 2, which is lower than pHpzc around 3 of the adsorbents.

R4. It is very hard to calculate the soprtion kinetics using rate equations. We need more tests done in order to get sufficient data in order to carry out the calculation.

R5. The results show that the adsorbents are not selective for the studied metals.

R6. It would very interesting to determine the influence of temperature on the sorption process. We think that sorption efficiency can decrease with increasing temperature because sorption is exothermic process. But we do not possess a suitable equipment to carry out this testing.

Round  3

Reviewer 2 Report

In Table 7, page 14 at the same pH-value (7.0) the adsorption capacity for micro/nano-structured Fe-Ni binary oxides amounts 90.1 mg/g with 245 m2/g specific surface area, but only 28,1; 3.3. and 1.5 mg/g with 298 m2/g, 274 m2/g and 253 m2/g for AliiI-coated leaching residue, FeII leached residue and FeIII leached residue, respectively. Can you explain it?

Conclusion: Can you give comparison between your results and Results from Referencse [32], [35], as previous mentioned in TAble 7..

This improved version after second revision and new offered answers can be accepted.

Author Response

Dear Reviewer,

Table 7 aims mainly to show that the modified adsorbents are effective as some nanoadsorbents, which were prepared using more complex procedures. The differences in sorption capacities for the mentioned adsorbents with the similar specific surface can be derived from their different chemical and mineralogical composition. The binary Fe-Ni oxides are crystalline while the modified adsorbents contain amorphous Al and Fe oxyhydroxides. The Al(III) oxyhydroxide probably attracts AS(V) anions stronger than Fe(II) and Fe(III) ones as well as Fe(II) over Fe(III) due to their bonding structures. Or the adsorbents can react with AS(V) anions, etc.

There can be many explanations for the obtained sorption results.At this stage the ambitions of the submitted manuscript  is not going into the details of sorption mechanisms of the adsorbents. We have been performing a study on sorption mechanisms in our recent research.